# Study protocol: Strengthening understanding of effective adherence strategies for first-line and second-line antiretroviral therapy (ART) in selected rural and urban communities in South Africa

Siphamandla Bonga Gumede[1,2]*, John Benjamin Frank de Wit[2], Willem Daniel Francois Venter[1], Samanta Tresha Lalla-Edward[1]

1 Ezintsha, Faculty of Health Sciences, University of Witwatersrand, Johannesburg, South Africa, 2 Faculty of Social and Behavioural Sciences, Department of Interdisciplinary Social Science, Utrecht University, The Netherlands

* sgumede@cartafrica.org, sgumede@ezintsha.org, s.b.z.gumede@uu.nl

**Funding:** The principal investigator/SBG is funded by the Utrecht University (UU), Department of

## Abstract

Multiple factors make adherence to antiretroviral therapy (ART) a complex process. This study aims to describe the barriers and facilitators to adherence for patients receiving first-line and second-line ART, identify different adherence strategies utilized and make recommendations for an improved adherence strategy. This mixed method parallel convergent study will be conducted in seven high volume public health facilities in Gauteng and one in Limpopo province in South Africa. The study consists of four phases; a retrospective secondary data analysis of a large cohort of patients on ART (using TIER.Net, an ART patient and data management system for recording and monitoring patients on ART and tuberculosis (TB)) from seven Johannesburg inner-city public health facilities (Gauteng province); a secondary data analysis of the Intensified Treatment Monitoring Accumulation (ITREMA) trial (a randomized control trial which ran from June 2015 to January 2019) conducted at the Ndlovu Medical Center (Limpopo province); in-depth interviews with people living with Human Immunodeficiency Virus (PLHIV) who are taking ART (in both urban and rural settings); and a systematic review of the impact of treatment adherence interventions for chronic conditions in sub-Saharan Africa. Data will be collected on demographics, socio-economic status, treatment support, retention in care status, disclosure, stigma, clinical markers (CD4 count and viral load (VL)), self-reported adherence information, intrapersonal, and interpersonal factors, community networks, and policy level factors. The systematic review will follow the Preferred Reporting Items for Systematic Reviews and Meta-Analysis (PRISMA) reporting and Population, Interventions, Comparisons and Outcomes (PICO) criteria. Analyses will involve tests of association (Chi-square and t-test), thematic analysis (deductive and inductive approaches) and network meta-analysis. Using an integrated multilevel socio-ecological framework this study will describe the factors associated with adherence for PLHIV who are taking first-line or second-line ART. Implementing evidence-based adherence approaches, when taken up, will improve patient's overall health outcomes. Our

Interdisciplinary Social Science, The Netherlands (https://www.uu.nl/en/organisation/faculty-of-social-and-behavioural-sciences), and by the Carnegie Corporation of New York (https://www.carnegie.org/) (Grant No–B 8606.R02), Swedish International Development Cooperation Agency (SIDA) (https://www.sida.se/en) (Grant No:54100113), the Developing Excellence in Leadership, Training and Science (DELTAS) Africa Initiative (https://www.aasciences.africa/aesa/programmes/developing-excellence-leadership-training-and-science-africa-deltas-africa#) (Grant No: 107768/Z/15/Z) and Deutscher Akademischer Austauschdienst (DAAD) (https://www.daad.de/en/). The DELTAS Africa Initiative is an independent funding scheme of the African Academy of Sciences (AAS)'s Alliance for Accelerating Excellence in Science in Africa (AESA) and supported by the New Partnership for Africa's Development Planning and Coordinating Agency (NEPAD Agency) with funding from the Wellcome Trust (UK) and the UK government. WDFV and STL-E are funded by the National Heart, Lung, And Blood Institute of the National Institutes of Health (https://www.nhlbi.nih.gov/) under Award Number UG3HL156388 and Fogarty International Centre. The content is solely the responsibility of the authors and does not necessarily represent the official views of the National Institutes of Health. The funders had and will not have a role in study design, data collection and analysis, decision to publish, or preparation of the manuscript.

**Competing interests:** The authors have declared that no competing interests exist.

study results will provide guidance regarding context-specific intervention strategies to improve ART adherence.

---

## Introduction

Inconsistent adherence to treatment is a contributing factor to poor health outcomes of people affected by numerous health conditions, including HIV, tuberculosis, diabetes mellitus (DM) and hypertension [1–3]. The World Health Organization (WHO) defines adherence as the degree to which a patient is able to follow a treatment schedule and take medication at recommended times [4–6]. In the context of HIV, lapses in adherence to medication can lead to the development of viral rebound, which can result in immunosuppression and viral resistance [4,7–9].

WHO recommends the use of the HIV drug dolutegravir (DTG) as the preferred first-line and second-line ART treatment regimens for all populations due to its formidable resistance barrier and improved tolerability [10–12]. Despite the advantages of this first-line regimen, in South Africa between 20%-30% of patients with HIV experience clinical, immunological or virological failure from first-line ART due to lapses in adherence [13–15]. This is a concern because of the clinical and cost implications attached to treatment failure [16,17].

Adherence to ART is a complex process that is affected by multiple factors, and numerous studies have attempted to establish what the barriers and facilitators of ART adherence are [18–20]. Individual-level factors such as age, sex, ethnicity, HIV status disclosure and forgetfulness, have been reported as important in predicting ART adherence [21]. However, using individual-level factors one is only able to report a limited proportion of the variability in non-adherence [22]. Good interpersonal relationships between patients and care givers or treatment supporters including healthcare providers, an intimate partner, family members, and friends have been reported as predictors for good adherence [22,23]. In contrast to intrapersonal and interpersonal factors, the community level factors such as poverty, HIV related stigma and discrimination against patients on ART introduce barriers to ART adherence [24]. Additional to community level factors, awareness of healthcare policy level factors like HIV treatment guidelines, policies, and best practices are imperative in ensuring good adherence and maintenance of the continuum of care [25].

## Conceptual framework

Various studies have demonstrated that there are many factors that play an important role in maintaining adherence behaviour [26–29]. These factors have been explored using several models, including: 1) Anderson's Health Care Utilisation model [30], which is a framework that considers predisposing factors (individual' own personality and behaviour), enabling factors (patient and health provider relationship, community education) and need factors (patient's beliefs, alternative medicine treatment options, community support); 2) the Dahlgren-Whitehead 'rainbow model' [31], a model that builds the relationship between the individual, the environment they live in and health; 3) Information-Motivation-Behavioural skills model (IMB model) [32–34], a model that views adherence behaviour as a function of the interrelations between adherence-related information, motivation, and behavioural skills; 4) the socio-ecological conceptual framework [35,36], which takes into consideration the individual, and their connections to other people, and how they adapt their behaviour to the social environment. This model suggests that an individual's behaviour is cohesive in a dynamic

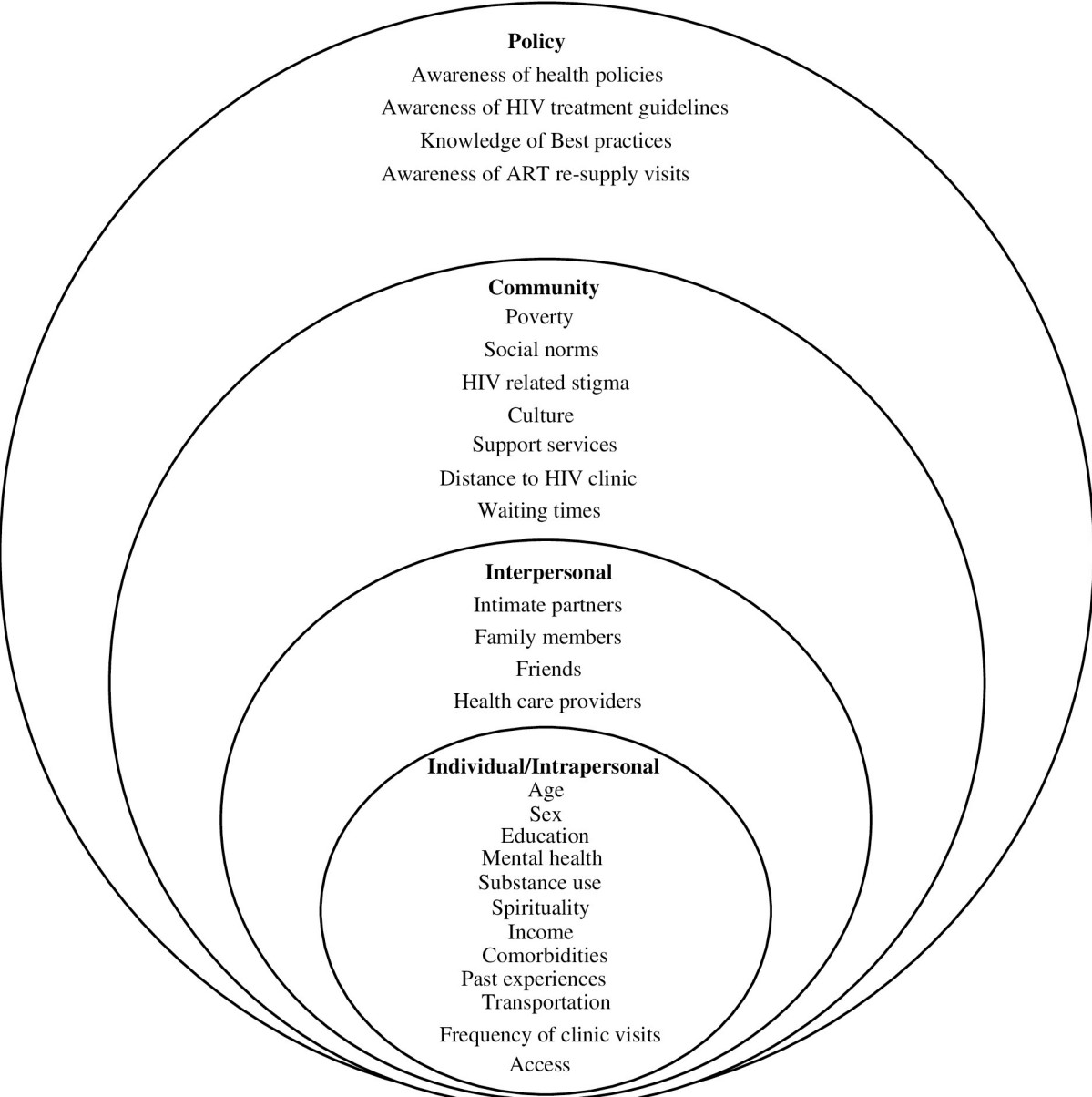

**Fig 1. An adapted socio-ecological framework.** An adapted socio-ecological framework that depicts the layers of individual, relationship, community, and healthcare policy level factors which influence the processes of treatment adherence and retention in care [36].

network of intrapersonal, interpersonal characteristics, community features and existing health policies [35,36].

Our study will adapt the socio-ecological conceptual framework to investigate multilevel and interactive factors such as individual/intrapersonal, interpersonal, community, and health policy level factors that affect adherence to ART (Fig 1). It will explore the different aspects of these factors that act either as barriers or facilitators of adherence to ART.

**Intrapersonal level factors.** The intrapersonal level of the socio-ecological conceptual framework comprises individual knowledge, attitudes, beliefs, perceptions, and skills that

influence behaviour [35]. It further includes age, sex, income, mental health, education, substance use, spirituality, comorbidities and past experiences [36].

**Interpersonal level factors.**   Here the participant's social network or relationships with other people including family, friends, peers, intimate partner and health care providers [35] are included. This level also highlights trust and communication as factors that builds a relationship between patients and care givers or treatment supporters [25].

**Community level factors.**   The components of the community-level factors that may influence patient's adherence to ART incorporate cultural views and social norms towards ART access, HIV related stigma, poverty, and available support services within the community [36].

**Policy level factors.**   The public policy level is shaped by local, and government laws regarding access and adherence to ART. The socio-ecological conceptual framework reports on awareness of macro level factors which includes health policies, HIV treatment guidelines and best practices, [22,36].

## Research gap

Although ample literature on adherence to ART has been available for a while, there remains a dearth of studies relating to the multi-level factors associated with adherence to treatment and processes shaping adherence behaviour, particularly in South Africa. This lends to the lack of understanding of the interplay between the various factors involved at different levels of the treatment taking behaviour of the ART patient.

In South Africa specifically, there is lack of reported knowledge about the effectiveness and impact of strategies currently employed to promote adherence in people living with HIV (PLHIV) who are taking ART (particularly young patients, males, and those experiencing severe treatment related side effects) [37,38]. Unfortunately, there are no quantitative studies of combination ART in South Africa (and even sub-Saharan Africa more generally) that have used a social-ecological perspective or approach. This is despite the understanding that patients on ART require comprehensive adherence strategies and strict viral load monitoring [39,40].

Until 2017, there were minimal related treatment adherence strategies (routine blood monitoring, targeted adherence counselling, and adherence clubs [41–43]) employed by the South African government [41]. However, effectiveness of these interventions have not been widely reported on. Additionally, the patients' perspectives seem to be given little consideration in the adherence intervention development and implementation. It is therefore important to assess effectiveness of existing strategies from the perspectives of PLHIV taking ART in order to recommend and develop relevant and acceptable strategies to patients [44,45]. This is evidenced in a study conducted in Malawi which highlighted how critical it is for health programmers to prioritize adherence view points from patients as part of strengthening the clinical monitoring strategy used [46].

## Rationale for the study

Consistent high levels of adherence to HIV medication are important for viral suppression, consequently preventing resistance to ART and progression of the illness [47]. According to the socio-ecological framework, numerous levels of factors affects patients' adherence to treatment [25]. Therefore, a multilevel socio-ecological framework will provide information about the influence of particular risk factors relative to others, or if combined effects of risk factors are additive or related. The socio-ecological framework will serve as the conceptual framework

in our study in order to understand factors affecting treatment adherence at different levels and also guide strategies to improve ART adherence.

**Aim.**    To describe the barriers and facilitators to adherence for patients receiving first-line and second-line ART and different adherence strategies utilised.

**Overarching hypothesis.**    A socio-ecological framework combined with multi-model data collection will identify the barriers and facilitators to ART adherence and retention in care for PLHIV needed to strengthen adherence interventions.

## Research questions and hypotheses

1.  What are the demographic characteristics and clinical indicators associated with virological failure and being lost to follow-up (LTFU) in PLHIV taking first-line and second-line ART in urban communities?
    Hypothesis: Demographic and clinical characteristics are associated with virological failure and being LTFU in patients on ART from complex, and highly mobile urban communities.

2.  What are the intra-and inter-personal factors (demographic, socio-economic characteristics), social and community level factors (poverty, social norms, HIV related stigma, culture), structural factors (health systems, support services) and clinical indicators associated with self reported adherence, pill count and virological failure to first-line and second-line ART in rural communities?
    Hypothesis: Individual and community level factors are associated with self-reported non-adherence, suboptimal pill count, and virological failure in patients on ART in the rural setting.

3.  What are the perspectives of virally suppressed and unsuppressed first-line and second-line ART patients about treatment adherence in selected urban and rural communities?

    a.  Are there any differences between people receiving first-line and second-line ART?

    b.  Are there any differences in treatment taking behaviours between virally suppressed and unsuppressed ART patients?

    c.  What do virally suppressed and unsuppressed ART patients recommend as adherence strategies?
        Hypothesis: In South African communities, treatment taking behaviour, perceptions of adherence, and recommendations for adherence interventions differ between PLHIV who are virally unsuppressed and those who are virally suppressed.

4.  What treatment adherence strategies and interventions have been implemented and evaluated in sub-Saharan Africa for HIV, hypertension, and DM?
    Hypothesis: There are no differences in the effectiveness of adherence interventions implemented in sub-Saharan Africa for HIV, hypertension, and DM.

## Materials and methods

### Study design

The study will employ a mixed method parallel convergent approach conducted in four phases (I-IV). Due to the different approaches for each study objective, we present the study methods based on each objective. While the studies will employ their own separate methods, all the results will be converged/collated into a single discussion to provide recommendations for adherence strategies that can be rolled out in South Africa.

## Materials and methods: Phase I

Objective 1: Assess demographic characteristics and clinical indicators associated with virological failure and LTFU in first-line and second-line patients in an urban community.

**Study design.** The study will employ a quantitative retrospective cohort study using secondary data analysis of data on people with HIV taking ART (18 years and older) recorded in the TIER.Net database.

TIER.Net is the monitoring and evaluation system used by the South African Department of Health for ART patient and data management. It comprises limited demographic information and all treatment and laboratory information from the time of commencement on HIV treatment. This is elaborated on in the data collection section.

The South African Department of Health started providing ART in the public health setting on 01 April 2004. From TIER.Net, we will extract a list of all patients who were initiated on ART from 01 April 2004 to 29 February 2020 in the urban setting (city of Johannesburg region F). The cut-off period of 29 February 2020 was chosen to give ART cohorts a minimum of a 12-month follow-up during which a full clinical assessment could be completed as per guidelines.

**Study setting.** This study will be conducted in seven health facilities in the city of Johannesburg region F.

This includes all levels of care

1. **Primary Health Care**: Jeppe Clinic, Malvern Clinic, Rosettenville Clinic, Yeoville Clinic

2. **Community Health Centre**: Hillbrow Community Health Center (HCHC)

3. **Hospitals**: Charlotte Maxeke Johannesburg Academic Hospital (CMJAH) and South Rand Hospital (SRH)

**Sampling/Sample size.** For this study, all records of people with HIV who were ever initiated on ART between 01 April 2004 (the inception of the South African national HIV treatment programme in the public health setting) and 29 February 2020 from the seven public health facilities will be included. Based on the TIER.Net database, about 130 000 adult patients were initiated on first-line and second-line ART in the seven facilities selected in the city of Johannesburg region F.

**Data collection.** Data will be extracted from the TIER.Net database (as an MS Excel export file). TIER.Net captures demographic information such as patient age, sex, facility name and contact details and also HIV specific information such as HIV diagnosis date, ART start date, regimen at baseline, ART visit dates, CD4 count, viral load done and viral load suppression The system also records all treatment related information, including ART switch. All these variables will be extracted (see S1 Appendix study codebook). The data will be exported to STATA 15.1 for data cleaning and analysis. Records with missing data will be excluded in the final analysis. This will be after conducting all necessary data quality checks, verifications, and triangulations with other data sources.

*Outcome.* VL count is categorized into suppressed (<1000 copies/ml) or unsuppressed (≥1000 copies/ml) [48,49]. The status on retention in care for patients will be categorized into active in care, LTFU, transferred out, or recorded dead. For this study, LTFU is defined as having missed a scheduled medical appointment by 90 days or more, as defined by the South African Department of Health [50].

**Data analysis.** Data will be coded and analysed using STATA version 15.1. Tests of association (Chi-square and t-test) between outcome variables and selected demographics

characteristics and clinic indicators will be conducted. Outcome variables will include viral load detectability and retention outcomes (active in care, transferred-out, lost to follow-up and dead),. Regression analysis, univariate and multivariate analyses between variables will be built for outcome variables to identify independent predictors. Variables such as age, sex, health facility, baseline regimen, ART start date, baseline CD4 count, most recent CD4 count, ART visit dates, and months on ART will be considered as predictor variables or independent variables. Analysis will include survival analysis which will consider different entry time points into the ART program. Subsequently, patients will not be grouped all together but will be followed up in a 12-month interval when measuring the outcomes (virological failure and lost to follow up).

## Materials and methods: Phase II

Objective 2: Assess socio-demographic and psychosocial associated with adherence (self-reported adherence, pill count and virological failure) in patients on first line and second-line ART in a rural community.

**Study design.** This cohort study using secondary data analysis will be conducted as a sub-study of the Intensified Treatment Monitoring Accumulation (ITREMA) study (Clinicaltrials.gov Identifier NCT03357588) [51]. The ITREMA database will be used to extract ART information. ITREMA is an open-label randomised clinical trial evaluating different treatment monitoring strategies for first-line ART, which ran from June 2015 to January 2019 [51]. ITREMA was conducted at the Ndlovu medical centre in Limpopo province, South Africa. The ITREMA study enrolled adult PLHIV and assessed an intensified HIV-treatment monitoring strategy in a randomised comparison with a control group receiving standard-of-care HIV treatment in accordance with the South Africa National Department of Health guidelines

**Study setting.** Patient enrolment for the ITREMA trial was done between June 2015 and August 2017 at the Ndlovu Medical Centre in Elandsdoorn, Limpopo Province, South Africa.

**Sampling/Sample size.** All the records from ITREMA database will be used. There are 501 ART patients in the ITREMA database.

**Data collection.** Data will be extracted from the ITREMA databases. The ITREMA database contains fields for socio-demographics and psychosocial characteristics. The control variables include sex, age, level of education, employment status, sources of income, household members, food security, mental health, HIV self-efficacy and HIV related stigma (see S1 Appendix study codebook and S6 Appendix ITREMA questionnaire). The data will be exported to STATA 15.1 for data cleaning and analysis.

*Outcome*. Self-reported non-adherence will be measured using three items from the ACTG questionnaire [52]: "How often do you have difficulty in taking your medication on time?, with responses given on 4-point scale (All the time, Most of the time, Rarely, Never), "On average how many days per week would you say that you missed at least one dose of your medication?", with responses given on a 6-point scale (every day, 4–6 days per week, 2–3 days per week, Once a week, Less than once a week, Never), and "When was the last time you missed taking any of your medications?", with responses also given on a 6-point scale (Past week, 1–2 weeks ago, 2–4 weeks ago, 1–3 months ago, More than 3 months ago, Never) (see S6 Appendix ITREMA questionnaire). Responding 'never' to all three questions will be taken to indicate good self-reported adherence. Suboptimal adherence measured using pill count is defined as a pill count <95%. This threshold is aligned with the WHO cut-off, which considers a pill count ≥ 95% as good adherence for patients taking ART [53]. Virological failure will be defined as viremia ≥1000 copies/ml within 96 weeks of follow-up [48,49].

**Data analysis.**    Data will be coded and analysed using STATA version 15.1. Tests of association (Chi-square and t-test) between outcome variables and selected socio-demographics and health related characteristics will be conducted. Outcome variables will include (but not limited to) viral load detectability, retention outcomes (active in care, transferred-out, lost to follow-up and dead), side effects and treatment interruptions (stop and restarting treatment). Regression analysis, univariate and multivariate analyses between variables will be built for outcome variables to identify independent predictors. Variables such as age, sex, health facility, beliefs, education, economic status, employment status, religious status, disclosure, and months on ART will be considered as predictor variables or independent variables.

## Materials and methods: Phase III

Objective 3: Understand adherence in first-line and second-line ART patients who are virologically suppressed and those who are not virologically suppressed in selected urban and rural communities.

**Study design.**    The study will employ a qualitative study design approach. Active patients from phase I and II (in both urban and rural settings) will be invited to participate in the in-depth interviews (IDIs) to explore factors including (but not limited to) treatment history, current use of ART, treatment regimen, financial/economic factors, risk behaviours (substance use), psychosocial characteristics cultural beliefs, spirituality and), relationship related factors (treatment support), community level factors (societal norms, stigma, discrimination, disclosure), and policy level factors (understanding/awareness of HIV/ART policies and treatment guidelines).

**Study setting.**    This study will be conducted in seven health facilities in the city of Johannesburg region F in Gauteng Province and one in Limpopo Province (Ndlovu Medical Centre).

The seven health facilities in the city of Johannesburg include:

1. **Primary Health Care**: Jeppe Clinic, Malvern Clinic, Rosettenville Clinic, Yeoville Clinic

2. **Community Health Centre**: Hillbrow Community Health Center (HCHC)

3. **Hospitals**: Charlotte Maxeke Johannesburg Academic Hospital (CMJAH) and South Rand Hospital (SRH)

**Sampling/Sample size.**    In this study, purposive sampling of patients currently taking ART will be employed to ensure maximum variation among the study sample. We will ensure a diverse sample by considering viral load status (suppressed and unsuppressed time on ART, age (18 years and older), sex (both males and females) and education. Sample size will depend on when saturation is reached; we anticipate that a maximum of 60 IDIs will be conducted across both study settings and viral load status groups (suppressed and unsuppressed). An anticipated number of study participants to recruit is 15 per viral load status group in each study setting (making a total of 30 participants in each study setting).

**Data collection.**    Patients for this phase will be contacted using the contact information that they provided for their facility records or databases used in phase I and II. IDIs will be conducted following a semi-structured interview guide. The guide will comprise open-ended questions covering treatment history, current use of ART and multilevel factors derived from the socio-ecological framework. This will include individual level factors including treatment-related factors (ART regimen, use of non-ART medication), financial and economic factors risk behaviours (substance use), psychosocial factors (cultural beliefs, spirituality),

interpersonal-level factors (relationship between patients and treatment supporters or caregivers, such as intimate partners, family members, friends and health care workers), community level factors (social norms regarding HIV and ART, HIV-related stigma and discrimination, HIV-status disclosure), health-system factors (access to HIV and ART services including adherence counselling), and policy level factors (HIV testing and treatment guidelines and policies). Additional probes will be included for each question to promote sharing of detailed information regarding their perspectives and experiences, and to ensure clarification if required (see S1 Appendix study codebook and S2 Appendix interview guide). All IDIs will be audio recorded and transcribed verbatim. Transcripts will be translated into English.

*Deductive themes.* Deductive themes may include treatment history, current use of ART, treatment regimen, financial/economic factors, risk behaviors (substance use), psychosocial characteristics cultural beliefs, spirituality and), relationship related factors (treatment support), community level factors (societal norms, stigma, discrimination, disclosure), and policy level factors (awareness of HIV/ART policies and treatment guidelines).

**Data analysis.**   Transcripts will be imported and analysed using NVIVO. Data coding will be undertaken using deductive (top down) and inductive approaches (bottom up) [54]. A deductive approach is driven by researchers' analytic interest in the study, reflecting the broad issues addressed in the interview guide. An inductive approach is used to identify the detailed themes related to the overarching issues that can be identified in the data [54]. Thematic analysis will be used, which is a method for identifying, analysing, and reporting patterns (themes) within data [54,55]. Transcripts will be read while noting similar topics that will be grouped into major topics or themes. Data will be analysed as transcripts become available shortly after interviews are conducted. This will ensure the early identification of emerging themes and assist in the identification of data saturation. Analysis of the IDIs will follow the phases of thematic analysis which are familiarization, generating initial codes, searching for themes, reviewing themes and interpretation [54]. Familiarization (getting grounded into the data collected), will be achieved by reading the transcripts and field notes repeatedly. During the process of generating codes, key emerging ideas and words from the familiarization phase will be recorded from which we will search, identify, and review themes, concepts, categories, and sub-categories. This will be done in keeping with socio-ecological framework, views and experiences that persist from the data. Finally, factors that influence adherence to ART will be identified and grouped into main categories. Analysis will be guided by the codebook which will be developed by the study team post familiarization with the data. There will be multiple independent coders to ensure the reliability of the coding.

## Materials and methods: Phase IV

Objective 4: Assess and compare adherence intervention strategies for the chronic conditions of HIV, hypertension and DM which have been tested and implemented in sub-Saharan Africa (Title: Adherence strategies and interventions for selected chronic conditions in sub-Saharan Africa: a systematic review and meta-analysis) (see S4 Appendix Systematic review protocol).

**Study design.**   This systematic review will be designed and reported according to the PRISMA [56] (see S5 Appendix PRISMA checklist), following the registered protocol (CRD42019127564) on the international prospective register of systematic reviews, Prospero [57] (see S4 Appendix Systematic review protocol). The study will use PICO criteria as the search strategy tool.

A systematic review on the impact of treatment adherence interventions in chronic conditions (HIV, hypertension, DM) in sub-Saharan Africa will be conducted to provide context to adherence in sub-Saharan Africa. In this region, HIV remains the leading cause of death more

especially in the young and middle-aged adults. However, the burden of non-communicable diseases (NCDs), particularly DM and hypertension, has increased rapidly in recent years [58–60].

**Study setting.**    All information from sub-Saharan Africa only will be included for the systematic review.

**Sampling/Sample size.**    All interventions described as chronic conditions adherence interventions (HIV/ART, hypertension, DM). Inclusion in the systematic review will be dependent on the criteria set out in the systematic review protocol [57] and reported using the PRISMA reporting guidelines.

**Data collection.**    A pre-defined data sheet will be developed for data extraction. The tool will include (but not be limited to): reference (author, title), year of publication, setting or location, sample size, intervention description, participants receiving adherence (in case of comparison) (see S1 Appendix study codebook). The form/tool will be tested before conducting the final searches. One reviewer will conduct all the data extraction while a second reviewer will be responsible for data quality assurance on the extraction and also conduct full text review of the included material.

We will search using several electronic databases. These will include PubMed/Medline, Web of Science, Google Scholar, Scopus, and CINAHL. If necessary, we will contact study authors and request more information on individual studies. Citations and bibliographies of records will be reviewed to identify additional relevant material.

The basic search terms included will be:

*"Chronic conditions"* OR *"hypertension"* OR *"high blood pressure"* OR *"blood pressure"* OR *"arterial hypertension"* OR *"mellitus diabetes type I"* OR *"mellitus diabetes type II"* OR *"Diabetes"* OR *"Sugar"* OR *"HIV"* OR *"Antiretroviral Therapy"* OR *"Antiretroviral Treatment"* OR *"ART"* OR *"ART Programs"* OR *"ART Programmes"* AND *"adherence"* OR *"compliance"* AND *"interventions"* OR *"strategies"* OR *"odds ratio"* OR *"risk ratio"* OR *"evaluation"* OR *"impact"* OR *"effectiveness"* OR *"outcome"* AND *"sub-Saharan Africa"* OR *"sub Saharan Africa"* OR *"sub-Saharan African"* OR *"sub Saharan African"* OR *"Africa"* (Table 1).

The search terms will be adjusted to suit the database being searched. An inventory with the database searched, the corresponding search criteria used, the date when the searches were conducted, and the results will be maintained. A second reviewer will run the searches separately for comparison. The strength of the body of evidence (quality of evidence), the risk of bias and magnitude of effect will be rated and assessed using Grading of Recommendations Assessment, Development and Evaluation (GRADE) [61,62].

*Outcome.* The primary outcome will be adherence to antiretroviral therapy, defined as the proportion of patients meeting the defined adherence criteria. The secondary outcome will be proportion of patients achieving viral suppression, as defined by the study. Outcome (and impact) measures will be reported in terms of changes in the prevalence or reduction in the relative risk. Should there be adequate statistical reporting, a meta-analysis will be considered [63].

**Data analysis.**    All adherence interventions or strategies will be described, based on the type of intervention implemented and the setting. The different evaluations methods will then be described in detail by comparing the type of assessments and outcome measures (adherence to ART and viral load). If appropriate, outcome measures will be reported in terms of changes in the prevalence or reduction in the relative risk. Whenever necessary, we will calculate unadjusted risk ratios (RRs) and 95% confidence intervals (CIs) from data provided and present the outcome indicator results in forest plots. Furthermore, we will perform a sensitivity analysis to measure the robustness of our results to the choice of summary statistic and calculated unadjusted risk differences. We will apply a random- effects model to calculate summary RRs and

**Table 1. Methodological aspect of the systematic review.**

| Criteria for study inclusion | Components details |
|---|---|
| Population (P) | Patients with selected chronic conditions (HIV, hypertension, dDM) in sub-Saharan Africa |
| Intervention (I) | All interventions listed/described as adherence interventions or strategies for the conditions of HIV, hypertension, DM |
| Comparisons (C) | Standard of care and other adherence interventions reported on in the review |
| Outcome (O) | The included studies should report any measurement of adherence to chronic conditions—primarily, effects on adherence behaviour and the changes in health outcomes. There is no preferred measurement for reporting; should there be adequate statistical reporting, a meta-analysis will be considered. |
| Setting | All studies from sub-Saharan Africa only will be considered for the review. |
| Language | There will be no language restrictions. |
| Date | There will be no date/time restrictions. |
| Publication status | All the documented studies will be considered and included for review. This includes peer reviewed (i.e., papers, manuscripts, and abstracts). |
| Method | The study will be designed and reported according to PRISMA. PICO will be used as a search strategy approach. This study will describe reported adherence programmes and strategies. There will be a focus on behaviour change techniques used or reported changes in process outcomes of adherence programmes and methods of implementation for HIV, hypertension and /or DM. |
| Search strategy and selection procedure | We will search using several electronic databases. These will include: PubMed/Medline, Scopus, CINAHL, Web of Science and Google Scholar. |
| Search terms | (chronic conditions OR hypertension OR high blood pressure OR blood pressure OR arterial hypertension OR mellitus diabetes type I OR mellitus diabetes type II OR Diabetes OR Sugar OR HIV OR Antiretroviral Therapy OR Antiretroviral Treatment OR ART OR ART Programs OR ART Programmes) AND (adherence OR compliance) AND (interventions OR strategies OR odds ratio OR risk ratio or evaluation OR impact OR effectiveness OR outcome) AND (sub-Saharan Africa OR sub Saharan Africa OR sub-Saharan African OR sub Saharan African OR Africa) |

95% Cl. To test the robustness of the findings, we will re-run the analysis using a fixed effects model. Data will be coded and analysed used STATA version 15.1.

Details for study criteria are presented in Table 1.

## Data management/Data cleaning (phase I-IV)

Data quality scripts for phase I, II, IV will be written in STATA. Data quality checks for phases I, II, and IV (quantitative data) will be done through RedCap (a secure web platform for building and managing research databases). Since TIER.Net will be used as the primary data source for phase I data extraction in the city of Johannesburg region F, where there are data quality issues, the study team will liaise with the facility staff and developmental partners in the region to assist with data clean-up activities. Data quality issues with the ITREMA data (phase II) will be communicated to the ITREMA study quality assurance officer for rectifying. Data quality checks for phase III (qualitative data) will be done in Microsoft Word (before exporting data to NVIVO). All transcripts will be checked for completeness and accuracy against original interviews. Creditability of data analysis will be ensured through triangulation of data sources (i.e., original interviews, field notes, transcripts, medical record).

## Data storage and access (phase I-IV)

Data will be captured and stored electronically, and password protected in the Microsoft Word, Microsoft Excel format and/or RedCap and will only be accessible to an investigator

and supervisors only. RedCap access is restricted to only those users who are registered on the system. All data from RedCap, Microsoft Word, Microsoft Excel, STATA, and voice files will be stored in access restricted folders on the Ezintsha server which will only be accessible to an investigator and supervisors. Any paper versions of data will be discarded after use. Data storage and access measures will also comply with data storage and access requirements of the Utrecht University.

## Ethical approvals and consent to participate

We obtained ethical clearance for all the phases of the study from the University of the Witwatersrand Human Research Ethics Committee (clearance certificate number: M190641). Departmental approval was granted by the Johannesburg Health District (DRC Ref: 2019-10-005 and National Health Research Database reference number: GP_201910_031). Written consent for interviews will be obtained from all participants. All participants will be provided with written information about the research, and they will also be verbally informed that their participation is voluntary and that they may withdraw from participation at any time (see S3 Appendix Participant Information sheet).

There are no risks or direct benefits for participating in the study. Participants may benefit by taking part in the study because many people find that it is useful to discuss their experiences, opinions and provide feedback. We believe that the information from this study will help the South African Department of Health better understand and strengthen the ART services provided to patients at large. All participants will be informed of these benefits.

All information discussed during the interview will be kept strictly confidential; at no point will participants' personal details be disclosed. The consent forms will be kept separately from all other research documents. Only the study team will have access to the information provided by the participants. All dissemination outputs will present de-identified and, where possible, aggregated data.

## Discussion

The overarching aim of this study is to contribute to knowledge that can provide guidance regarding the barriers and facilitators to adherence for first-line and second-line ART patients and different adherence strategies utilized. Using an integrated multilevel socio-ecological framework, this study will focus on determining the influence of the multiple factors that impact on adherence to ART. In line with socio-ecological frameworks [64,65] and propositions, the findings from this study will be discussed under four units of analysis: intrapersonal level, interpersonal level, community level and policy level factors.

### Intrapersonal factors

The client knowledge, attitudes, experiences and perceptions coupled with analysis of the intrapersonal factors influencing adherence to ART play a fundamental role in maintaining adherence [35]. Intrapersonal level factors such as being age, sex, substance abuse, and comorbidities will be discussed, in line with the existing literature [66–69].

### Interpersonal level factors

The trust and communication between patients on ART and treatment supporters is essential in improving and maintaining optimal adherence to treatment [25]. Some studies have reported treatment support as a predictor of adherence [38,70]. In this study, the interpersonal level factors consisting of relationships between patients and family members, friends, intimate

partner(s) and healthcare providers will be discussed against previous studies to provide guidance on the role of treatment supporters in strengthening treatment adherence.

## Community level factors

The findings concerning community level factors will consider the importance of patients understanding of social norms, cultural barriers and reduction of poverty, stigma and discrimination against people living with HIV and on ART [71]. Information on poverty, culture, HIV related stigma, and discrimination caused by misconceptions will be discussed in line with existing literature to provide a critical assessment on the role of community level factors on ART adherence.

## Policy level factors

The policy-level factors address awareness and influence of public health policies, guidelines, and standards on patients [72,73]. The South African HIV programme has undergone several changes since its implementation in 2004 [74]. This study will evaluate patient understanding/ awareness of ART adherence or HIV treatment related health policies, guidelines, and best practices that are followed by health providers when providing health services. Knowledge of ART medicines or regimens (names of ART drugs), definitions, and clinical functions of viral load (knowledge of threshold for virological failure and suppression) and CD4 cell count (understanding of high or low CD4 cell count) will be discussed as themes in assessing individual's awareness of existing HIV treatment policies and guidelines. Additionally, we will be able to provide recommendations on required changes at the policy level to improve treatment adherence.

**Strengths and limitations.**   To our knowledge this will be the first study using a social-ecological perspective and a convergent parallel mixed method design to report on combination ART in South Africa. Combining quantitative and qualitative data from public health settings, a controlled environment, the patient perspective and evidence from the region will enable us to make recommendations for a comprehensive, acceptable, and appropriate adherence strategy for the country.

The study presents a fewlimitations. The study will be conducted in a total of eight health facilities (seven of over 120 health facilities in one South African metropolitan municipality (urban setting) and one facility in a rural setting). Therefore, findings may not be generalizable to other municipalities and districts in South Africa, or to other country settings. Furthermore, although there are efforts to ensure good quality of data by the South African Department of Health, supporting partners and research staff, secondary data are subject to quality issues, due to data inconsistencies and missing data. Other potential predictors of adherence, such as disclosure, stigma, and self-reported adherence, will be assessed through a standard questionnaire. This may lead to reporting bias (memory and social desirability biases). Policy-level factors might not probe for relevant factors due to how this study is designed (with it focus on the individual's awareness of health policies and guidelines which is insufficient to fully assess policy-level factors).

In conclusion, our study will demonstrate how an existing socio-ecological conceptual framework can be used as a tool to provide guidance regarding facilitators and barriers to ART adherence. By populating this framework through secondary data analysis, participant interviews of PLHIV who are taking ART and a systematic review comparing adherence intervention strategies for the chronic conditions, this mixed method study will provide evidence on factors affecting treatment adherence at different socio-ecological levels and guide context-specific intervention strategies to improve ART adherence. We believe that the use of our

study results to strengthen adherence intervention will subsequently improve health outcomes and decrease the number of patients switching to complex treatment such as second-line and third-line regimens.

## Dissemination

Findings from this research will be submitted for doctoral degree purposes (by thesis). Peer reviewed publications and scientific conference presentations will be developed. Results of the research will be shared with the research participants, donors, and health facilities.

## Supporting information

**S1 Appendix. Study codebook.**
(PDF)

**S2 Appendix. Interview guide.**
(PDF)

**S3 Appendix. Participant information sheet.**
(PDF)

**S4 Appendix. Systematic review protocol.**
(PDF)

**S5 Appendix. PRISMA checklist.**
(PDF)

**S6 Appendix. ITREMA questionnaire.**
(PDF)

## Acknowledgments

This research is supported by the Consortium for Advanced Research Training in Africa (CARTA). CARTA is jointly led by the African Population and Health Research Center and the University of the Witwatersrand. The statements and views made in this article are solely the responsibility of the authors.

We would like to thank all the relevant health and research authorities from the City of Johannesburg and Ndlovu Medical Center for allowing the research team to engage in a partnership to strengthen health service delivery through technical assistance and research.

## Author Contributions

**Conceptualization:** Siphamandla Bonga Gumede, John Benjamin Frank de Wit, Willem Daniel Francois Venter, Samanta Tresha Lalla-Edward.

**Funding acquisition:** Siphamandla Bonga Gumede, John Benjamin Frank de Wit, Willem Daniel Francois Venter, Samanta Tresha Lalla-Edward.

**Methodology:** Siphamandla Bonga Gumede, John Benjamin Frank de Wit, Willem Daniel Francois Venter, Samanta Tresha Lalla-Edward.

**Supervision:** John Benjamin Frank de Wit, Willem Daniel Francois Venter, Samanta Tresha Lalla-Edward.

**Writing – original draft:** Siphamandla Bonga Gumede, John Benjamin Frank de Wit, Samanta Tresha Lalla-Edward.

**Writing – review & editing:** Siphamandla Bonga Gumede, John Benjamin Frank de Wit, Willem Daniel Francois Venter, Samanta Tresha Lalla-Edward.

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
