## [Decision Letter · Decision Letter 0]

19 Oct 2021

PONE-D-21-13178Study protocol: Strengthening understanding of effective adherence strategies for first-line and second-line antiretroviral therapy (ART) in selected rural and urban communities in South AfricaPLOS ONE

Dear Dr. Siphamandla Bonga Gumede,

Thank you for submitting your manuscript to PLOS ONE. After careful consideration, we feel that it has merit but does not fully meet PLOS ONE’s publication criteria as it currently stands. Therefore, we invite you to submit a revised version of the manuscript that addresses the points raised during the review process.

ACADEMIC EDITOR:

We look forward to receiving your revised manuscript.

Kind regards,

Obinna Ikechukwu Ekwunife, PhD

Academic Editor

PLOS ONE

Journal Requirements:

Reviewers' comments:

Reviewer's Responses to Questions

**Comments to the Author**

1. Does the manuscript provide a valid rationale for the proposed study, with clearly identified and justified research questions?

Reviewer #1: Partly

Reviewer #2: Partly

2. Is the protocol technically sound and planned in a manner that will lead to a meaningful outcome and allow testing the stated hypotheses?

Reviewer #1: Yes

Reviewer #2: Partly

3. Is the methodology feasible and described in sufficient detail to allow the work to be replicable?

Reviewer #1: Yes

Reviewer #2: No

4. Have the authors described where all data underlying the findings will be made available when the study is complete?

Reviewer #1: Yes

Reviewer #2: Yes

5. Is the manuscript presented in an intelligible fashion and written in standard English?

Reviewer #1: Yes

Reviewer #2: Yes

6. Review Comments to the Author

You may also provide optional suggestions and comments to authors that they might find helpful in planning their study.

Reviewer #1: The authors present a clear and detailed study protocol a complex study on ART adherence. The planned study will address four distinct research questions, each using four distinct analytic approaches including a routine healthcare record analysis, a secondary analysis of clinical trial data, interviews with patients, and a systematic review of literature.

Given the complexity of the study, it would be helpful to explain why each component is needed and how results from the different components will come together to address the overarching study aims. It would be especially helpful if the researchers could provide an overarching central hypothesis and hypotheses for each of the research questions, and then comment on how the components will come together to address the overarching aims of the study.

There are a few points of confusion with the conceptual framework and how the study components will use the framework.

First, the framework and the analysis do not include access-related enablers such as distance to clinic/pharmacy, wait time in clinic/pharmacy, frequency of clinical visits and ART re-supply visits, etc. These have been shown to influence ART medication possession and adherence. How will these be accounted for in the framework and analysis?

Second, medication reimbursement policies are included in the framework, but will any study participants be exposed to requirements for any form of payment for ART? How will this be evaluated, given that ART medications are available free-of-cost in public sector clinics in South Africa?

Third, the way in which the researchers plan to elicit policy-level factors seems to be probing not the policy-level factors themselves (which have a direct effect on adherence) but rather an individual’s understanding of health policies and guidelines. These seem to be two different things and it is not clear that one can be substituted for the other. For example, a patient may not be familiar with what the current policies are, or the names of ART medications, but they may still be affected by policies, for example, by being prescribed those medications, whether or not they know their names.

Finally, in the Introduction, the following claim is unsubstantiated and contrasts with the earlier discussion of numerous extant frameworks for understanding adherence. “Unfortunately, most adherence studies are not guided by any conceptual framework and subsequently are unable to draw adherence strategies from conceptual frameworks of any kind.”

Appendices are thorough and detailed. Phase 2 codebook is missing a data dictionary/glossary for the variable abbreviations.

The paper has a number of minor grammatical problems and need a thorough proof-read for sentence structure and punctuation consistency. A few grammatical issues:

Abstract: open parenthesis before “(using” is never closed.

Introduction:

“adherences” – I’m not sure this can be plural

“However, individual-level factors are only able to report a limited portion of the variability in non-adherence.” – here the subject ‘individual-level factors’ is not doing the verb ‘report.’ Also ‘portion’ → proportion?

“Good interpersonal relationships between patients and care givers or treatment supporters including healthcare providers, an intimate partner, family members, and friends have been reported as a predictor for good adherence.” – should make ‘a predictor’ plural

“intrapersonal, interpersonal” – word repeated

Reviewer #2: The protocol submitted by Gumede et al proposes 4 parallel studies to answer questions about barriers and facilitators of adherence and useful interventions to support adherence. This field of research is not new, but conclusive results are not yet available therefore new evidence is welcome.

Despite the ambitious plan of conducting a four phases study and the use of a predefined conceptual framework, I am not sure that the proposed methodology allows to reach the expectations. In addition there is little explanation about how mixed methods will be applied.

The research gap is not well documented by an in-depth analysis of the abundant literature available on the subject and falls short of building a strong justification (rational) to the study.

Research questions: it is not clear to me the difference between the first and the second question if not for the setting (urban vs rural) and the used dataset (cohort vs RCT). Or is it that the first question uses virological failure and the second adherence as main outcome? but in data analysis the selected outcomes are the same.

Material and methods:

Phase 1: the possible sample size for this study is very interesting, but the type of dataset is very limited. this should be discussed as well as how missing data will be managed (missing data are very frequent in routinely collected data). Another important point to discuss is the time frame of the analysis. Will all patients be evaluated at the 12-month follow up? adherence being dynamic, possible barriers and facilitators will be very different for someone after 1 year of treatment or after 17 years despite similar characteristics at baseline. how survival bias will be managed? (the less adherent may have died over time) How this point will be managed? Some of the control variable seem not available in the list provided (code-book)

Phase 2: this study is a secondary analysis of data collected during an RCT. The trial should be presented to understand the population involved and the type of intervention evaluated to allow the reader to understand if the study can answer the question. Nowhere is defined how adherence is measured and the codebook provided is unintelligible to the non initiated to the database. Again the outcome variables are ill defined and do not include adherence (see research question)

Phase 3: I think this may be the most fruitful phase of the study as it stands on primarily collected data.

The other comment on this phase is that I feel that even if interviewed patients can provide their perspectives on health system and policy factors, the research to be multilevel should include perspectives from the other "levels" included in the framework, being them service providers or policy makers.

Phase 4: I do not understand why the study protocol for this phase is provided in supplementary material instead of in the main body of the protocol. I am not experienced in protocols for systematic review and meta-analysis, but my impression is that neither in the main protocol nor in the one provided in the supplementary material the study methodology is very clear. What is the difference between interventions and strategies for the authors? will they compare single adherence support activities or categories of intervention (Cognitive behavioral therapy? incentives? Behavioral skills training? etc)

Will they consider only clinical trials? how will they manage the quality of the study selected? How will they deal with the different measurement of outcomes? I feel that there is already a lot of work done on adherence and if the methodology of review is not rigorous also in terms of choice of outcome and way of measuring, the study will contribute only to more confusion. on this point I advise to include in the bibliography Lancet HIV 2017; 4: e31–40

Concerning availability of data after the study, very little is discussed; with a generic "data ca be made available on reasonable request". I understand that more and more it is requested that data are made public on specific repository. this should be discussed.

Concerning ethical aspects: in this paragraphe it is usually requested to discuss benefices and risks for participants and the measures taken to safeguard confidentiality and privacy of participants. The obtained ethical clearance concerns all the phases of the protocol?

Discussion: readers are expecting in this chapter the discussion of the strengths and limits of the study, how the study fits with the multilevel socio-ecological framework, and how the parallel mixed method design will enrich interpretation of the data. In fact what is missing is the mixing of the methods: the protocol as presented it is more a multi-studies protocol, not indicating how the mixing of methods will be done.

Minor comments:

- update the paragraphe about WHO first line recommandations to include dolutegravir

- check that all abbreviations are given first in extent

- it is useful to define a single primary outcome to make the statistics, with the other endpoints being secondary

- it is useful to give the definitions used for each variable and its categories: in the supplementary material it is not very clear for the study phase 2

7. PLOS authors have the option to publish the peer review history of their article (what does this mean?). If published, this will include your full peer review and any attached files.

Reviewer #1: **Yes: **Anna Bershteyn

Reviewer #2: No

---

## [Author Response · Author response to Decision Letter 0]

28 Oct 2021

As requested by the Journal staff, I have provided two revised manuscript (labelled as Revised Manuscript with Track Changes and Manuscript), a letter of response to reviewers and new supporting information labelled as: S7 Appendix_ITREMA questionnaire

Please do not hesitate to contact me if you require more information

---

## [Decision Letter · Decision Letter 1]

16 Nov 2021

PONE-D-21-13178R1Study protocol: Strengthening understanding of effective adherence strategies for first-line and second-line antiretroviral therapy (ART) in selected rural and urban communities in South AfricaPLOS ONE

Dear Dr. Siphamandla Bonga Gumede,

Thank you for submitting your manuscript to PLOS ONE. After careful consideration, we feel that it has merit but does not fully meet PLOS ONE’s publication criteria as it currently stands. Therefore, we invite you to submit a revised version of the manuscript that addresses the points raised during the review process.

We look forward to receiving your revised manuscript.

Kind regards,

Obinna Ikechukwu Ekwunife, PhD

Academic Editor

PLOS ONE

Journal Requirements:

Reviewers' comments:

Reviewer's Responses to Questions

**Comments to the Author**

1. Does the manuscript provide a valid rationale for the proposed study, with clearly identified and justified research questions?

Reviewer #1: Partly

Reviewer #2: Yes

2. Is the protocol technically sound and planned in a manner that will lead to a meaningful outcome and allow testing the stated hypotheses?

Reviewer #1: Yes

Reviewer #2: Yes

3. Is the methodology feasible and described in sufficient detail to allow the work to be replicable?

Reviewer #1: Yes

Reviewer #2: Yes

4. Have the authors described where all data underlying the findings will be made available when the study is complete?

Reviewer #1: Yes

Reviewer #2: Yes

5. Is the manuscript presented in an intelligible fashion and written in standard English?

Reviewer #1: Yes

Reviewer #2: Yes

6. Review Comments to the Author

You may also provide optional suggestions and comments to authors that they might find helpful in planning their study.

Reviewer #1: Now that I understand this paper is for a student's degree requirement, I recommend that the student iterate the hypotheses with their academic advisor in order to choose hypotheses based on gaps in the field. The hypothesis that "PLHIV who have optimum adherence to ART and remain in care will have better treatment and clinical outcomes" does not appear to be a gap in the field because it is well-known than HIV care and adherence to ART improves clinical outcomes and indeed saves lives of PLHIV.

The critique regarding inappropriate methods for evaluating policy-level factors was not adequately addressed. I recommend changing the study aims/framing to be evaluating comprehension/awareness of policy, rather than evaluating the effect of policy on patient outcomes, which this study is not currently designed to do.

Typo: "asherence"

Reviewer #2: The authors have considered reviewers' comments and amended the protocol consequently. The overarching hypothesis is not of the most in need of confirmation, but the final results will be of interest for the global knowledge on adherence and for the local authorities for policy making.

As the authors state at different sections of their answer, the protocol is for a doctoral thesis and have to comply with different requirements. I congratulate the authors for their major endeavor in realizing all these studies.

7. PLOS authors have the option to publish the peer review history of their article (what does this mean?). If published, this will include your full peer review and any attached files.

Reviewer #1: No

Reviewer #2: No

---

## [Author Response · Author response to Decision Letter 1]

21 Nov 2021

This is a submission of our revised manuscript, accompanied by the following documents (uploaded)

• A rebuttal letter that responds to each point raised by the academic editor and reviewer(s). The letter is labelled 'Response to Reviewers'.

• A marked-up copy of our manuscript that highlights changes made. This file/document is labelled 'Revised Manuscript with Track Changes'.

• An unmarked version of our revised paper without tracked changes. This file/document is labelled 'Manuscript'.

In addition, we have uploaded an updated figure 1 (an adapted socio-ecological framework)

---

## [Editor Report · Decision Letter 2]

25 Nov 2021

Study protocol: Strengthening understanding of effective adherence strategies for first-line and second-line antiretroviral therapy (ART) in selected rural and urban communities in South Africa

PONE-D-21-13178R2

Dear Dr. Siphamandla Bonga Gumede,

We’re pleased to inform you that your manuscript has been judged scientifically suitable for publication and will be formally accepted for publication once it meets all outstanding technical requirements.

Kind regards,

Obinna Ikechukwu Ekwunife, PhD

Academic Editor

PLOS ONE
---

## [Editor Report · Acceptance letter]

13 Dec 2021

PONE-D-21-13178R2 

Study protocol: Strengthening understanding of effective adherence strategies for first-line and second-line antiretroviral therapy (ART) in selected rural and urban communities in South Africa 

Dear Dr. Gumede:

I'm pleased to inform you that your manuscript has been deemed suitable for publication in PLOS ONE. Congratulations! Your manuscript is now with our production department. 

Kind regards, 

on behalf of

Dr. Obinna Ikechukwu Ekwunife 

Academic Editor

PLOS ONE